Drug susceptibility and molecular epidemiology of Escherichia coli in bloodstream infections in Shanxi, China

Zhang Yanjun 1
Wang Hairu 2
Li Yanfang 1
Hou Yabin 3
Hao Chonghua teacherhaochonghua@163.com 1
1 Department of Clinical Laboratory, Shanxi Provincial People’s Hospital, Affiliate of Shanxi Medical University , Taiyuan , China
2 Department of Clinical Laboratory, Shanxi Bethune Hospital , Taiyuan , China
3 Department of Clinical Laboratory Diagnostics, Shanxi Medical University , Taiyuan , China
Mora-Montes Hector
Electronic publication date: 2021 Oct 25
Publication date: 2021
Volume: 9
Electronic Location ID: e12371
Received 2021 Jun 15; Accepted 2021 Oct 1
Copyright: ©2021 Zhang et al.
Copyright year: 2021
Copyright holder: Zhang et al.
License: This is an open access article distributed under the terms of the Creative Commons Attribution License, which permits unrestricted use, distribution, reproduction and adaptation in any medium and for any purpose provided that it is properly attributed. For attribution, the original author(s), title, publication source (PeerJ) and either DOI or URL of the article must be cited.
License URL: https://creativecommons.org/licenses/by/4.0/

Keywords: Escherichia coli, Drug susceptibility, Bloodstream infections, Molecular epidemiology, Extended-spectrum beta-lactamases (ESBLs)

Funding: The funders received no funding for this work.

==============================
Objectives

We carried out a retrospective study to investigate the drug susceptibility and genetic relationship of clinical Escherichia coli isolates from patients with BSIs in Shanxi, China.

Methods

E. coli isolates causing BSIs were consecutively collected from June 2019 to March 2020. Antimicrobial susceptibility testing was performed by broth microdilution method. PCR was used to detect antimicrobial resistance genes coding for extended-spectrum β-lactamases (ESBLs), phylogenetic groups and seven housekeeping genes of E. coli.

Results

A total of 76 E. coli were collected. Antimicrobial susceptibility testing revealed that the top six E. coli resistant antibiotics were ampicillin (90.7%), ciprofloxacin (69.7%), cefazolin (65.7%), levofloxacin (63.1%), ceftriaxone and cefotaxime (56.5%). Among the 76 isolates, 43 produced ESBLs. Molecular analysis showed that CTX-M-14 was the most common ESBLs, followed by CTX-M-15 and CTX-M-55. Phylogenetic group D (42.2%) predominated, followed by group B2 (34.2%), group A (18.4%) and group B1 (5.2%). The most prevalent sequence types (STs) were ST131 (15/76), ST69 (12/76) and ST38 (6/76).

Conclusions

This study is the first to report the phenotypic and molecular characteristics of E. coli isolated from BSIs in Shanxi, China. Our results indicated a high prevalence of MDR in E. coli strains isolated from BSIs and a serious spread of ESBL genes in Shanxi, especially the epidemiological blaCTX-M. Phylogenetic analysis indicated genetic diversity among E. coli BSIs isolates.

Introduction

Bloodstream infections (BSIs), one of the most common severe infections, is also an important complication leading to extended hospitalization and increased mortality (Yoon et al., 2018). BSIs is occurred in about 2 million episodes and is responsible for about 250000 deaths in United States and Europe, ranking among the top seven causes of death in North America and Europe (Goto & Al-Hasan, 2013; Sader et al., 2019).

In recent years, the main etiology of BSIs has been patterns of antimicrobial resistance, especially with Gram-negative bacteria (Sader et al., 2019). Escherichia coli (E. coli) is an important causative pathogen of Gram-negative BSIs, whether in view of healthcare-associated infection surveillance or antimicrobial resistance in modern global health (Kern & Rieg, 2019; Tsuzuki et al., 2020; Xiao et al., 2019a). The incidence of BSIs caused by E. coli has increased in Europe, with an annual increase of 6% between April 2012 and March 2014 (Bou-Antoun et al., 2016). Data from the SENTRY Antimicrobial Surveillance Program showed that the proportion of E. coli in all BSIs isolates increased from 18.7% in 1997–2000, to 24.0% in 2013–2016 (Diekema et al., 2019). The most frequently organism identified from blood samples, was E. coli (23.1%) according to the statistical results from the China antimicrobial surveillance network in 2018 (Hu et al., 2019).

The 30-day all-cause mortality rate after E. coli BSIs is about 16%, which may increase with the rises of antimicrobial resistance (Vihta et al., 2018). Extended-spectrum β-lactamases (ESBLs) production is the main drug resistance mechanism identified in E. coli (Nepal et al., 2017). Unfortunately, Tian, Zhang & Sun (2019) found that the proportion of E. coli producing ESBLs increased significantly from 0% in 1998–2002 to 76.2% in 2008–2012 in China. The increase of the acquired multidrug resistance rate of ESBLs-producing E. coli has placed significant restrictions on appropriate and reliable antimicrobial therapeutic options (Bi et al., 2017). Timely administration of appropriate empiric antimicrobial therapy can save lives, but the most appropriate empiric antimicrobial regimen requires a full understanding of the common causes of BSIs and its drug resistance patterns (Marchello et al., 2019). Data on full-scale susceptibility surveillance and molecular epidemiological investigation of E. coli causing BSIs in Shanxi Province of China are lacking. In this article, we set out to analyze the susceptibility of antimicrobial agents, distribution of drug resistance genes, genetic relationship and sequence types (ST) of E. coli causing BSIs in Shanxi, China.

Materials & Methods

Setting and study design

This retrospective study was conducted in a general teaching hospital affiliated to Shanxi Medical University (Shanxi Provincial People’s Hospital) in Taiyuan, the capital city of Shanxi Province in North China. Shanxi Provincial People’s Hospital is a comprehensive tertiary hospital, with 2,800 beds and a total of approximately 1.28 million patients annually. The hospital provides better medical services to the people of Shanxi Province, about 370,000 people every year.

The study was approved by the Ethical Committee of Shanxi Provincial People’s Hospital, Shanxi Medical University (Ethical Application Ref: 2020-7). Written informed consent was obtained from participants for our study.

Bacterial isolates

We performed a retrospective cross-sectional study of patients with E. coli BSIs between June 2019 and March 2020. Cases of E. coli BSIs were identified from the laboratory database in the Department of Clinical Laboratory, Shanxi Provincial People’s Hospital. Each patient was included only once, if multiple blood cultures from the same patient were positive, only the first episode was reviewed and recorded. All isolates were identified by matrix-assisted laser desorption ionization-time of flight mass spectrometer (bioMérieux, Marcy-l’Étoile, France) and stored in broth containing 30% glycerol at −80 °C until further experiments.

Antimicrobial susceptibility testing and detection of ESBLs and carbapenemases

Antimicrobial susceptibility testing was carried out by broth microdilution method according to Clinical and Laboratory Standard Institute (CLSI) recommendations (CLSI, 2019) for ampicillin, amoxicillin-clavulanate, piperacillin-tazobactam, cefazolin, cefoxitin, ceftriaxone, cefepime, cefotaxime, cefoperazone-sulbactam, aztreonam, ertapenem, imipenem, meropenem, amikacin, gentamicin, tobramycin, ciprofloxacin, levofloxacin, tigecycline and trimethoprim-sulfamethoxazole. The breakpoint of tigecycline was based on European Committee on Antimicrobial Susceptibility Testing (EUCAST) (EUCAST, 2019), other drugs were interpreted by CLSI and cefoperazone/sulbactam was referred to cefoperazone in CLSI (CLSI, 2019). E. coli ATCC 25922 and ATCC 35218 were used as quality control strains for antimicrobial susceptibility testing.

Broth dilution test (cefotaxime and cefotaxime-clavulanic acid, ceftazidime and ceftazidime-clavulanic acid) was used as confirmatory test for ESBLs producers following CLSI criteria (CLSI, 2019). Either cefotaxime or cefotadime combined with clavulanic acid, the MIC was decreased by threefold concentration, which can be considered as ESBLs positive. Klebsiella pneumoniae ATCC 700603 was used as positive control for ESBLs production. According to the susceptibility results, the ESBLs genes were further detected in the positive isolates.

Detection of resistance genes

Template DNA was extracted by boiling method as mentioned in previous article (Zhao et al., 2015). Polymerase chain reaction (PCR) were performed to screen for blaTEM, blaSHV, blaCTX-M(−1,−2,−8,−9,−25 group), blaV EB, blaGES, blaOXA(−1,−2,−10 group), and blaPER genes, using reverse and forward primers pairs listed in Table 1, as previously described (Wang et al., 2016). Positive amplicons were sequenced using ABI3730xlDNAAnalyzer by Sangon Biotech (Shanghai, China) and the DNA sequences were compared in GenBank (http://blast.ncbi.nlm.nih.gov) to identify the allelic variant.

Table 1 Sequences of primers for resistance genes PCR amplification.

Gene	Primersa	Primer sequences (5′-3′)	Expected amplicon size (bp)	
TEM	F
R	ATAAAATTCTTGAAGACGAAA
GACAGTTACCAATGCTTAATC	1,080	
SHV	F
R	TGGTTATGCGTTATATTCGCC
GGTTAGCGTTGCCAGTGCT	865	
CTX-M-1	F
R	AAAAATCACTGCGCCAGTTC
AGCTTATTCATCGCCACGTT	415	
CTX-M-2	F
R	CGACGCTACCCCTGCTATT
CCAGCGTCAGATTTTTCAGG	552	
CTX-M-8	F
R	TCGCGTTAAGCGGATGATGC
AACCCACGATGTGGGTAGC	666	
CTX-M-9	F
R	CAAAGAGAGTGCAACGGATG
ATTGGAAAGCGTTCATCACC	205	
CTX-M-25	F
R	GCACGATGACATTCGGG
AACCCACGATGTGGGTAGC	327	
OXA-1	F
R	CTGTTGTTTGGGTTTCGCAAG
CTTGGCTTTTATGCTTGATG	440	
OXA-2	F
R	CAGGCGCYGTTCGYGATGAGTT
GCCYTCTATCCAGTAATCGCC	233	
OXA-10	F	GTCTTTCRAGTACGGCATTA	822	
	R	GATTTTCTTAGCGGCAACTTA		
VEB	F
R	GCGGTAATTTAACCAGA
GCCTATGAGCCAGTGTTC	961	
GES	F
R	ATGCGCTTCATTCACGCAC
CTATTTGTCCGTGCTCAGG	846	
PER	F
R	AGTCAGCGGCTTAGATA
CGTATGAAAAGGACAATC	978	
Notes.

a Primer, The ‘F’ meant the forward primer and the ‘R’ meant the reverse primer.

Phylogenetic group analysis and multilocus sequence typing

According to the well recognized phylogenetic grouping protocol proposed by Clermont, Bonacorsi & Bingen (2000), four major phylogenetic groups (A, B1, B2 and D) in the E. coli isolates were determined using the method of triple PCR based on three genetic markers, namely chuA, yjaA and TspE4.C2. chuA encodes outer membrane hemin receptor gene that involves in heme transport. yjaA encodes for gene responsible for cellular response to hydrogen peroxide and acid stress and TspE4.C2 DNA encodes for putative lipase esterase gene (Javed, Mirani & Pirzada, 2021). PCR was performed to determine the seven conserved housekeeping genes (adk, fumC, gyrB, icd, mdh, purA, and recA) of E. coli, and the primer pairs were listed in Table 2. The allelic profiles and STs were described by the combination of the seven alleles on line (http://mlst.warwick.ac.uk/mlst/dbs/Ecoli/).

Table 2 Primers sequences of the housekeeping genes of E. coli.

Gene	Primersa	Primer sequences (5′-3′)	Expected amplicon size (bp)	
adK	F	ATTCTGCTTGGCGCTCCGGG	536	
	R	CCGTCAACTTTCGCGTATTT		
fumC	F	TCACAGGTCGCCAGCGCTTC	469	
	R	TCCCGGCAGATAAGCTGTGG		
gyrB	F	TCGGCGACACGGATGACGGC	460	
	R	GTCCATGTAGGCGTTCAGGG		
icd	F	ATGGAAAGTAAAGTAGTTGTTCCGGCACA	518	
	R	GGACGCAGCAGGATCTGTT		
mdh	F	AGCGCGTTCTGTTCAAATGC	452	
	R	CAGGTTCAGAACTCTCTCTGT		
purA	F	TCGGTAACGGTGTTGTGCTG	478	
	R	CATACGGTAAGCCACGCAGA		
recA	F	CGCATTCGCTTTACCCTGACC	510	
	R	AGCGTGAAGGTAAAACCTGTG		
Notes.

a Primer, The ‘F’ meant the forward primer and the ‘R’ meant the reverse primer.

Statistical analysis

The statistical analysis was performed using SPSS 25.0 (IBM, Armonk, NY, USA). When fitting a normal distribution, we presented continuous numerical variables as mean and standard deviation (SD). For categorical variables, results were expressed as the percentages of the groups from which they were derived. The χ2 test was used to compare categorical variables. All tests were two-sided, and p < 0.05 was considered statistically significant.

Results

Characteristics of total patient population

A total of 76 eligible E. coli isolates were enrolled during the study period. From the total of 76 E. coli causing BSIs they were isolates from females (42/76) and (34/76) from males. The age of patients ranged from 13 to 90 years and the mean age was 62.1 ± 14.9 years. The mean age did not differ significantly in ESBLs-producing isolates compared to non-ESBLs-producing E. coli.

Antimicrobial susceptibility tests

Antimicrobial susceptibility testing revealed that the top 6 E. coli resistant antibiotics were ampicillin (90.7%), ciprofloxacin (69.7%), cefazolin (65.7%), levofloxacin (63.1%), ceftriaxone and cefotaxime (56.5%). On the contrary, carbapenems, piperacillin-tazobactam, amikacin and tigecycline exhibited excellent activity against E.coli isolates in vitro with susceptibility up to 100.0%. Of the 76 isolates, 56.5% were confirmed as ESBLs-producing E. coli. Compared with ESBLs-producing E. coli, the non-ESBLs-producing E. coli showed higher susceptibility to cefazolin, ceftriaxone, cefepime, cefotaxime, aztreonam (P < 0.05) (Table 3). Despite the significant differences, both ESBLs-producing and non-ESBLs-producing E. coli showed lower susceptibility to ampicillin, ciprofloxacin, levofloxacin, trimethoprim-sulfamethoxazole and gentamicin. Out of total 76 E. coli isolates, 56 (73.6%) were multidrug resistant (MDR) (nonsusceptibility to ≥1 agent in ≥3 antimicrobial categories) isolates. The rates of MDR isolates were 97.6% and 42.4% in ESBL-producing and non-ESBL-producing E. coli, respectively.

Table 3 Rates of antimicrobial resistance among E. coli bloodstream isolates.

Antimicrobial agents	Number of isolates (%)	p	
	Total (n = 76)	ESBLs (n = 43)	non-ESBLs (n = 33)		
AMP	69 (90.7)	43 (100.0)	26 (78.7)	0.002	
AMC	5 (6.5)	4 (9.3)	1 (3.0)	0.274	
TZP	0	0	0	–	
CFZ	50 (65.7)	43 (100.0)	7 (21.2)	<0.001	
FOX	11 (14.4)	9 (20.9)	2 (6.0)	0.068	
CRO	43 (56.5)	43 (100.0)	0	<0.001	
CEF	11 (14.4)	11 (25.5)	0	0.002	
CTX	43 (56.5)	43 (100.0)	0	<0.001	
CPS	1 (1.3)	1 (2.3)	0	0.378	
ATM	28 (36.8)	28 (65.1)	0	<0.001	
ETP	0	0	0	–	
IPM	0	0	0	–	
MEM	0	0	0	–	
AK	0	0	0	–	
GM	40 (52.6)	26 (60.4)	14 (42.4)	0.118	
TOB	3 (3.9)	3 (6.9)	0	0.122	
CIP	53 (69.7)	37 (86.0)	16 (48.4)	<0.001	
LEV	48 (63.1)	36 (83.7)	12 (36.3)	<0.0001	
TGC	0	0	0	–	
SXT	38 (50.0)	23 (53.4)	15 (45.5)	0.488	
Notes.

AMP, ampicillin; AMC, Amoxicillin-clavulanate potassium; TZP, piperacillin–tazobactam; CFZ, Cefazolin; FOX, Cefoxitin; CRO, ceftriaxone; CEF, cefepime; CTX, Cefotaxime; CPS, Cefoperazone-sulbactam; ATM, aztreonam; ETP, ertapenem; IPM, imipenem; MEM, meropenem; AK, amikacin; GM, gentamicin; TOB, tobramycin; CIP, ciprofloxacin; LEV, levofloxacin; TGC, Tigecycline; SXT, trimethoprim–sulfamethoxazole.

Characterization of resistance genes

A total of 71 drug-resistant genes including 28 blaCTX-M-14, 20 blaTEM-1, 10 blaCTX-M-15, 7 blaOXA-1 and 6 blaCTX-M-55 were identified among the 43 ESBLs-producing isolates, where they have one, two and three ESBLs genes in 19 isolates 20 and 4 isolates respectively. The highest detection rate was blaCTX-M-14 (39.4%, 28/71), followed by blaTEM-1 (28.2%, 20/71), blaCTX-M-15 (14.1%, 10/71), blaOXA-1 (9.8%, 7/71) and blaCTX-M-55 (8.5%, 6/71) (Table 3). No blaSHV, blaCTX-M (−2,−8,−25 group), blaGES, blaV EB, blaOXA (−2,−10 group) or blaPER genes were found.

Phylogenetic characterization

The phylogenetic lineages were determined in the 76 isolates: phylogenetic group D (42.2%) predominated, followed by group B2 (34.2%), group A (18.4%) and group B1 (5.2%). Among the 43 ESBLs-producers, phylogenetic group D (41.8%) predominated, followed by group B2 (37.3%), A (16.2%) and B1 (4.7%). On the other side, among the 33 non-ESBLs-producers, phylogenetic group D (42.4%) predominated, followed by group B2 (30.3%), A (21.2%) and B1 (6.1%). The percentage of phylo-groups identified were the same regardless of the ESBLs producing. Among the 40 blaCTX-M positive strains, phylogenetic group B2 (40.0%) predominated, followed by group D (37.5%), group A (17.5%) and group B1 (5.0%) (Fig. 1).

Figure 1 Distribution of phylogenetic groups.

Multilocus sequence typing

Among the 76 isolates, a total of 28 different sequence types (STs) were identified in this study (Table 4). The ST131 was the most frequently ST identified (19.7%, 15/76), followed by ST69 (15.7%, 12/76), ST38 (7.8%, 6/76), ST1193 (6.5%, 5/76), ST648 (5.2%, 4/76), ST73 (5.2%, 4/76), ST46 (3.9%, 3/76), ST405 (3.9%, 3/76) and other uncommon STs that were detected in one or two isolates. Among ESBLs-producing isolates, the most prevalent STs were ST131 (27.9%, 12/43), ST69 (9.3%, 4/43) and ST38 (9.3%, 4/43), while among non-ESBLs-producers, the predominant STs were ST69 (24.2%, 8/33), ST73 (12.1%, 4/33) and ST131 (9.0%, 3/33). As shown in Table 4, all ST131 assigned to phylogenetic group B2, harbored the TEM-1 (6/20), CTX-M-14 (11/28), CTX-M-15 (3/10), CTX-M-55 (1/6) and OXA-1 (3/7). All ST69 and ST38 assigned to phylogenetic group D, while, 3 (3/12) ST69 isolates and 4 (4/6) ST38 isolates were ESBL-producers.

Table 4 Phylogenetic groups and genotypes in MLST of 76 E. coli isolates.

Sequence Type	Phylogenetic groups	Resistance determinants	No. of isolates	
ST131	B2	TEM-1, CTX-M-14	5	
	B2	CTX-M-14	3	
	B2	–	3	
	B2	CTX-M-15, CTX-M-14, OXA-1	2	
	B2	CTX-M-55, CTX-M-14	1	
	B2	TEM-1, CTX-M-15, OXA-1	1	
ST69	D	–	8	
	D	TEM-1	3	
	D	CTX-M-14	1	
ST38	D	–	2	
	D	CTX-M-14	1	
	D	CTX-M-15	1	
	D	TEM-1, CTX-M-14	1	
	D	CTX-M-15, CTX-M-14	1	
ST1193	B2	–	2	
	B2	CTX-M-14	1	
	B2	TEM-1, CTX-M-55	1	
	B2	TEM-1, CTX-M-15	1	
ST73	B2	–	4	
ST648	D	–	2	
	D	CTX-M-14	1	
	D	CTX-M-55	1	
ST46	A	CTX-M-14	2	
	A	CTX-M-55	1	
ST405	D	TEM-1, CTX-M-14	3	
ST117	D	TEM-1, CTX-M-14	1	
	D	TEM-1, CTX-M-55	1	
ST450	A	–	2	
ST2003	D	TEM-1, CTX-M-14	1	
	D	CTX-M-14	1	
ST2448	B1	–	2	
ST44	A	CTX-M-15, OXA-1	1	
ST10	A	–	1	
ST410	A	TEM-1, CTX-M-15, OXA-1	1	
ST457	D	TEM-1, CTX-M-15	1	
ST2179	B1	CTX-M-14, OXA-1	1	
ST393	D	–	1	
ST1163	D	–	1	
ST155	B1	CTX-M-14	1	
ST773	A	–	1	
ST1284	A	CTX-M-15, OXA-1	1	
ST95	B2	–	1	
ST493	B2	CTX-M-55	1	
ST542	A	–	1	
ST409	A	–	1	
ST167	A	–	1	
ST4503	A	CTX-M-14	1	

Discussion

The increased consumption of antimicrobial agents, the high prevalence and dissemination of drug resistance genes in pathogens, and the poor prevention and control strategies for infections lead to the increase of antimicrobial resistance (Parajuli et al., 2017). Approximately 700,000 deaths in antimicrobial resistance every year in the world, which is expected to soar to a staggering 10 million in 2050 (Huh et al., 2020). As pathogens’antimicrobial resistance rates and mortality in BSI patients increase, monitoring of microorganisms and antimicrobial resistance has become critical (Mehl et al., 2017). Our study not only monitored the resistance phenotype of E. coli, but also detected the distribution of drug resistance genes and the genetic relationship of the isolates, which provided the basis for designing strategies for the treatment and prevention of these serious infections.

The present study showed that resistance of several commonly used antibiotics used to treat BSIs in our area, such as penicillins, cefazolin, ceftriaxone, cefotaxime, fluoroquinolones and folate pathway inhibitors, was high (Table 3), which was consistent with previous report (Zhao et al., 2015). Fortunately, carbapenems, piperacillin-tazobactam, amikacin and tigecycline have low drug resistance rates, which should be considered for empirical treatment of E. coli isolates in our region. In this study, MDR accounted for 73.6% in E. coli causing BSIs, the value is lower than the result (85.6%) of another similar study in China by Ma et al. (2017), but much higher than that in E. coli causing pyelonephritis (40%), sepsis (32%) and skin and soft tissue infections (26%) (Ranjan et al., 2017). Our study showed that among the 43 ESBLs-producing E. coli, 97.7% were MDR, while among 33 non-ESBLs-producing, only 42.4% were MDR. Since the ESBLs genes are usually found in large plasmids which also contain other antimicrobial resistant genes, most ESBLs producing organisms are MDR isolates (Ma et al., 2017).

In this study, the proportion of ESBLs-producing E. coli is 56.5%, which was consistent with that in Zhejiang (57.6%) (Xiao et al., 2019b), but much higher than in Japan (26.1%), Vietnam (39.3%), Singapore (33%), and Brazil (12.8%) (Hung et al., 2019; Komatsu et al., 2018). Since the 2000s, CTX-M have replaced TEM as the most common type of ESBLs in many countries (Liao et al., 2017; Zou et al., 2019). In the present study, genotypes showed remarkable increase in the CTX-M (40/71) compared to TEM (20/71) in ESBLs-producing E. coli. It is noteworthy that blaCTX-M were strongly associated with ST131 (80% carried blaCTX-M). We found that blaCTX-M-14 was the predominant blaCTX-M, followed by blaCTX-M-15 and blaCTX-M-55, while in Germany, blaCTX-M-15 was the most common and blaCTX-M-14 was the second most frequently identified genotype (Rohde et al., 2020). CTX-M-55, a variant of CTX-M-15 that contains a substitution of A80V within the β-lactamase possessing enhanced cephalosporin-hydrolyzing activity, was rarely found in clinical isolates previously (Liao et al., 2017; Zhang et al., 2019). While, in some surveys, the prevalence of blaCTX-M-55 has surpassed blaCTX-M-15 and become the secondary genotype of blaCTX-M (Hu et al., 2018; Zhao et al., 2015). It was observed that the predominant of subtypes blaCTX-M gene may change with region and time. In addition, isolates producing blaOXA-1 also accounted for a larger proportion among the ESBLs-producers and mainly belonged to B2-ST131.

In this study we identified 28 STs among 76 E. coli isolates and MLST showed abundant genetic diversity in the E. coli. Similar to other studies in the world (Van Hout et al., 2020; Yasir et al., 2018), ST131 was the predominant ST. Moreover, we found that the predominant ESBLs type in the B2-ST131 (phylogenetic group-sequence type) isolates was CTX-M-14, rather than CTX-M-15 as reported in Australia (Harris et al., 2018), indicating an association of distinct CTX-M types with different settings due to various modes of transmission. In our study, ST69 was the most prevalent among non-ESBLs-producers, while one study from Netherlands showed that ST73 was the most common sequence type among non-ESBLs-producing isolates (Van Hout et al., 2020). This proved that the distribution of STs of E. coli vary significantly by region. Our study found that males accounted for 60.0% of patients infected with ST131 and females accounted for 83.3% of patients infected with ST69, consistent with a Canadian study (Holland et al., 2020). The reasons for genders differences in E. coli ST types are unclear and require further study.

Conclusions

In conclusion, data for this article described the drug sensitivity and molecular epidemiology of E. coli in patients with BSIs in Shanxi, China. We found that E. coli were highly resistant to commonly used antibiotics, except for carbapenems, piperacillin-tazobactam, amikacin and tigecycline. The B2-ST131 and D-ST69 clonal groups were the most common clinically relevant genotypes. Phylogenetic analysis showed genetic diversity among E. coli isolates. Better monitoring of the epidemiology of E. coli bacteremia is needed to develop and implement effective prevention strategies.

Supplemental Information

Supplemental Information 1 Raw data

Click here for additional data file.

We would like to thank Yu Zhang of Shanxi Medical University for her help in our experiment.

Additional Information and Declarations

Competing Interests

Author Contributions

Ethics

Data Availability

The authors declare there are no competing interests.

Yanjun Zhang and Hairu Wang performed the experiments, authored or reviewed drafts of the paper, and approved the final draft.

Yanfang Li and Yabin Hou analyzed the data, prepared figures and/or tables, and approved the final draft.

Chonghua Hao conceived and designed the experiments, authored or reviewed drafts of the paper, and approved the final draft.

The following information was supplied relating to ethical approvals (i.e., approving body and any reference numbers):

The study was approved by the Ethical Committee of Shanxi Provincial People’s Hospital, Shanxi Medical University (Ethical Application Ref: 2020-7).

The following information was supplied regarding data availability:

The raw data are available in the Supplementary File.

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
