# Peer review of "Drug susceptibility and molecular epidemiology of Escherichia coli in bloodstream infections in Shanxi, China"

_PeerJ, doi:10.7717/peerj.12371_

## Round 0.1 · original submission · Major Revisions

Please pay particular attention to the concerns raised about the experimental design.

Reviewer 1 ·

Basic reporting

In general, the research is carried out with scientific rigor, and the objectives set are achieved using the methods that were used.

Experimental design

In general, the methods used, as well as their description, are acceptable. However, I will make some suggestions in order to improve the work.

Line 112. What do the ESBL genes refer to, for example, blatTEM, etc? for better understanding I suggest defining what those genes are or placing a table that defines them

Line 120. E. coli italic font.

Line 121. What do the chuA and yjaA genes code for?

Line 157. What are those 71 drug-resistant genes? Mention if it is in the supplementary material.

Line 206-207. The method with which you isolate genomic DNA, also allows the maintenance of plasmid DNA for later analysis?

In the Bacterial Isolates section, I recommend describing the process of isolating the pathogen and not only the identification process or reference to this method.

In Table 3, for ease of use, I suggest placing the abbreviations of each of the antibiotics at the bottom of the table.

In figure 2, what do the colors mean?
What are those housekeeping genes of E.coli?
Or mention them in the phylogenetic group analysis and multilocus sequence typing section.

Validity of the findings

The conclusions of the research are well supported by the results obtained. And I believe that the contributions of this study will help in the correct medication to treat the BSIs caused by this pathogen.

235-237. Is there any hypothesis why this difference between genders?

Additional comments

In general, the research is carried out in a good way and the results are medically important. Consider the above suggestions.

Reviewer 2 ·

Basic reporting

The manuscript by Zhang and colleagues describes the molecular epidemiology of Escherichia coli in bloodstream infection. The manuscript lacks the description of phylogenetic background (the details of genetic relationship by phylo-groups and the goeburts analysis will be different to the real mean of phylogenetic background as the study of evolutionary relationships of species or genes). Minor re-organization of the manuscript is required, together with inclusion of additional details to fully validate the conclusions drawn. I have a few comments related to the manuscript that I would like to pass on to the authors for consideration and revision.

Scientific Points
Will be include the province of china hospitals in the title – Drug susceptibility and molecular epidemiology of Escherichia coli in bloodstream infections in Shanxi China.
1. Line 2. Correct the word susceptiblity.
2. Line 20-22. This sentences is a brief introduction, the objective is describe in line 22-24, modify the order.
3. Line 23. Phylogenetic background will be refer as genetic relationship.
4. Line 30-32. The antimicrobial susceptibility in the raw-data, do not show high rates to the ampicillin, ciprofloxacin, cefazolin, levofloxacin, ceftriaxone and cefotaxime. Explain what’s the mean of (>56%)?
5. Line 38. Our results indicated that the antimicrobial resistance rates were high. The conclusion will be modified with deep comparison and additional details, explain why you affirm the existence of “high rates of antimicrobial resistance”.

6. Line 54. Delete year by year.

7. Line 57-58. The most frequently organism identified from blood samples, was E. coli (23.1%) according to the statistical results from the China antimicrobial surveillance network in 2018.

8. Line 62. …resistance mechanism identified in E.coli…

9. Line 63. Lei Tian et al, found…

10. Line 72. Phylogeny will be change by genetic relationship.

11. Line 103. Cefotaidime?

12. Line 105. Isolate resistant to any of the antibiotics ertapenem, meropenem or imipenem was considered as carbapenemase-producer.

13. Line 114. The primers sequences is different from the reference Du et al. 2014.

14. Line 116. Change gene subtype for allelic variant.

15. Line 126. This is a copy-page from the (https://www.phyloviz.net/goeburst/) “using a graphic matroid approach that ensures an optimal solution for the placement of links between Sequence Types”

16. Line 129. Use the original reference of Feil et al. 2004.

17. Line 138. From the total of 76 E. coli causing BSIs they were isolates from females (42/76) and (34/76) from males.

18. Line 148. The non-ESBLs-producing E. coli showed higher susceptibility to cefazolin, ceftriaxone, cefepime, cefotaxime, aztreonam (P<0.05) (Table 2).

19. Line 152-155. Rephrase the sentences to avoid misunderstandings.

20. Line 157. A total of 71 drug-resistant genes were identified among the 43 ESBLs-producing isolates, where they have one, two and three ESBLs genes in 19 isolate 20 and 4 isolates respectively.

21. Line 164. The phylogenetic lineages were determined in the 76 isolates:

22. Line 169. The percentage of phylo-groups identified were the same regardless of the ESBL producing.

23. Line 172. …and group B1 (5.0%) (Figure 1).

24. Line 175. …were identified in this study (Table 3)

25. Line 175. The ST131 was the most frequently ST identified (19.7%, 15/76), followed…

26. Line 180. Change “dominated” by predominant

27. Line 181. As shown in Table 3, all ST131 assigned to phylogenetic group B2, harbored the TEM-1 (6/20),

28. Line 196. The asseveration of “highly resistant to penicillins, cefazolin, etc” need to be supported with examples and values.

31. Line 225. This sentence will be changed, the phylo-group and goeBurst analysis only shows the genetic relationship between the isolates and does not give any clue about their evolution. ”E. coli causing BSIs did not evolve from a unique ancestral background”

32. Line 243. Need to rephrase the sentence according to the analysis.

Experimental design

29. Line 205. From the 33 non-ESBLs-producing, 28 were resistance to Ampicillin how do you explain it if you didn’t identify any of the principals Beta-lactamase?

30. Line 205. At less 48 of the isolates were resistant to Lev and 53 resistant to Cip. Why you didn’t explore the mechanisms to quinolone resistance in order to explain in part the MDR in the non-ESBLs-producing.

33. Figure 2. The goeBurst analysis show the allelic relationship between the different ST. The analysis with the option of TLV, show that ST44 and ST167 are at DLV from the ST10. The rest of the ST did not have real genetic relationship between each other’s. I suggest eliminating figure 2 or doing the analysis with the complete ST database of E. coli. In order to give a complete picture of the ST found in Shanxi China with respect to the main CC identified in the world.
ST: 10
Profile: 10 11 4 8 8 8 2 10
ST: 44
Profile: 10 11 4 8 8 8 7 44
ST: 167
Profile: 10 11 4 8 8 13 2 167
ST: 1284
Profile: 10 4 4 8 8 13 73 1284

Validity of the findings

No comments

Additional comments

No comment

---

## Round 0.2 · Minor Revisions

One reviewer suggested minor changes to the manuscript. Please address them accordingly.

Reviewer 1 ·

Basic reporting

No comment

Experimental design

No comment

Validity of the findings

No comment

Additional comments

Thank you very much for taking into account my comments and suggestions, which have been with the aim of improving the manuscript.

Reviewer 2 ·

Basic reporting

In general, the report is consistent with the objectives, methodology and conclusion. The authors' modifications based on the suggestions have improved the understanding of the document.

Experimental design

The methods are acceptable and were improved.

Validity of the findings

Line 128. "the algorithm was set to group strains together if 6 of 7 alleles were homologous as most studies"
I do not agree with the aseveration that the relationship between isolates can be made in a study, using 6 of the 7 alleles of the MLST of E.coli. In most research studies, the authors use at least 4 identical alleles to consider that they have a genetic relationship.

Line 173. "(group 1: ST45, ST4503;" modified the ST45 for the ST405

In figure 2 modified the ST405 in the text

In figure 2, I don't understand how the ST10 is related to the rest of the STs. Goeburts Full MLST analysis, with level 6 option, does not show ST10.

Additional comments

It is necessary to explain the relevance of figure 2; otherwise it should be removed from the report.

Annotated reviews are not available for download in order to protect the identity of reviewers who chose to remain anonymous.

---

## Round 0.3 · accepted · Accept

The manuscript was again improved following the reviewers' suggestion and is now suitable for publication in Peer Journal.